 elife.elifesciences.org

# Evolution of thorax architecture in ant castes highlights trade-off between flight and ground behaviors

**Roberto A Keller[1,2]\*, Christian Peeters[2], Patrícia Beldade[1]**

[1]Instituto Gulbenkian de Ciência, Oeiras, Portugal; [2]Laboratoire Écologie & Évolution, CNRS UMR 7625, Université Pierre et Marie Curie, Paris, France

**Abstract** The concerted evolution of morphological and behavioral specializations has compelling examples in ant castes. Unique to ants is a marked divergence between winged queens and wingless workers, but morphological specializations for behaviors on the ground have been overlooked. We analyzed thorax morphology of queens and workers in species from 21 of the 25 ant subfamilies. We uncovered unique skeletomuscular modifications in workers that presumably increase power and flexibility of head–thorax articulation, emphasizing that workers are not simply wingless versions of queens. We also identified two distinct types of queens and showed repeated evolutionary associations with strategies of colony foundation. Solitary founding queens that hunt have a more worker-like thorax. Our results reveal that ants invest in the relative size of thorax segments according to their tasks. Versatility of head movements allows for better manipulation of food and objects, which arguably contributed to the ants' ecological and evolutionary success.

## Introduction

A detailed understanding of morphology is of prime importance to elucidate how organisms evolved and operate in nature. This is especially so in an era of increasingly sophisticated developmental genetic analysis, as the correct interpretation of molecular data depends largely on the precise characterization of morphological structures (e.g., *Prud'homme et al., 2011* vs *Yoshizawa, 2011*; *Mikó et al., 2012*). During development, the relative investment in the growth of different body parts (e.g., allocation of nutritional resources to somatic vs germ tissues) will determine adult morphologies, and thus influence an organism's ecology (*Nijhout and Emlen, 1998*; *Emlen, 2001*). Morphology interacts very closely with behavior in shaping phenotypic evolution (*Baldwin, 1896*; *Simpson, 1953*; *Robinson and Dukas, 1999*). On the one hand, changes in behavior will often influence the environment in which organisms are selected, leading to modifications of morphology (*Wcislo, 1989*; *Crispo, 2007*). On the other hand, morphological specializations can open the potential for further behavioral change (*West-Eberhard, 2003*).

Specializations that associate morphology and behavior have compelling examples in insect polyphenisms, where alternative morphologies result from environmental regulation of development and are typically associated with distinct behavioral repertoires (*Beldade et al., 2011*; *Simpson et al., 2012*). For example, horned and hornless male beetles produced as a result of nutritional plasticity have different reproductive tactics (guarding vs sneaking access to females in nests [*Moczek and Emlen, 2000*]). In many social insects, differential feeding leads to the production of distinct queen and worker castes, each with characteristic morphology and behavior underlying reproductive vs maintenance functions within the colony (*Wheeler, 1986*; *Beldade et al., 2011*), and increasing colony performance as a whole (*Oster and Wilson, 1978*).

Among the social Hymenoptera, ants are an extreme case of caste polyphenism, because queens are usually winged and workers are always wingless (*Wilson, 1971*; *Hölldobler and Wilson, 1990*). Flight allows queens to disperse from the natal nests before they start new colonies, while the lack of

**\*For correspondence:** roberto.kellerperez@gmail.com

**Competing interests:** The authors declare that no competing interests exist.

**Reviewing editor**: Diethard Tautz, Max Planck Institute for Evolutionary Biology, Germany

**eLife digest** The size and shape of an animal, known as its morphology, often reflect the actions it can perform. A grasshopper's long legs, for example, are well suited to hopping, whilst the streamlined body of a dolphin helps swimming through water. These specialized features result from the interplay between morphology and behavior during evolution. A change in morphology can make new behaviors possible, which can then expose the animal to new environments and selective pressures that, in turn, can lead to further changes in morphology.

The interplay between morphology and behavior is particularly interesting in social insects such as ants. Queens and workers within an ant colony have a similar set of genes, but they have dramatically different morphologies and very different roles within the colony. Queens are responsible for reproduction, and are larger and have wings, which allow them to fly and establish a new colony away from where they were born. Workers are smaller and lack wings, and they devote themselves to building the nest, feeding the young larvae and protecting the colony. This marked morphological divergence, unique to ants, has fascinated researchers for more than a century. However, most studies have focused on the presence or absence of wings and have overlooked the interactions between morphology and the actions performed on the ground.

Like all insects, an ant's body is divided into three parts: the head, the thorax (to which the legs and wings are attached), and the abdomen. Now, Keller et al. have examined the shape of the thorax in many species of ants and found that workers are not just smaller wingless versions of queens: rather, the architecture of their thorax is unique among species of flying insects. The front end of the worker thorax is greatly enlarged and is filled by strong neck muscles that power the head and its jaws, and allow workers to hunt and carry prey many times their own weight.

Keller et al. also identified two distinct types of queens and went on to show that these two shapes evolved in association with the two types of strategy that lone queens use to found new colonies. In species where queens convert their own wing muscles into the food for the first generation of workers, the wing muscles are much enlarged and the neck segment is extremely reduced. In species where queens hunt to feed the new colony, the wing and neck muscles are more balanced in size. As such, for those ant species where very little is known about how new colonies are founded, Keller et al. show that we can use the shape of the queen's thorax to help predict this behavior.

Taken together, the results of Keller et al. show that female ants invest in the relative size of the different segments of the thorax in a way that reflects their behavior as adults. These adaptations partly explain why ants have been so extraordinarily successful in nature, and underscore the importance of carefully analyzing an organism's form to fully understand its biology.

wings in workers is thought to facilitate the exploitation of ground habitats and cramped spaces (**Hölldobler and Wilson, 1990**). The presence and operation of wings is tightly associated with the morphology of the thorax. In the typical thorax of modern flying insects, the first segment (T1) bears no dorsal appendages, while the second (T2) and third (T3) each bear a pair of wings (**Snodgrass, 1935**). Because of this, studies of morphological specializations of the insect thorax have focused on the wing-bearing segments T2 and T3. The relative size of these segments varies widely across insect orders, but tends to be conserved within (**Dudley, 2002**). Surprisingly, the entire thoracic skeletomuscular architecture of ant castes, including the T1 segment that forms the articulation with the head, has been neglected, from both functional and comparative perspectives.

In this study, we use a phylogenetically broad comparative approach, involving queens and workers from species representing 21 of the 25 extinct and extant ant subfamilies, to investigate external morphology and internal anatomy in the context of caste-specific specialized behaviors. Our analysis reveals a unique modification of the thoracic architecture in worker ants, presumably connected with their powerful head and mandibles, and uncovers two types of thoracic configurations in queens, associated to different strategies for the foundation of new colonies.

## Results

To characterize caste-specific modifications in the architecture of the thorax, we first quantified the length of the thoracic segments in queens and workers of 11 species and performed anatomical

dissections in multiple individuals from 19 species, representing eight ant subfamilies. We unveiled a unique thorax architecture in workers vs queens and then confirmed the generality of our findings in an extended sample of species across ant diversity. The quantitative analysis of thorax morphology showed that queens fall in two distinct anatomical types. Using parsimony and maximum likelihood (ML) methods, we reconstructed the pattern of thoracic evolution onto the established phylogenetic tree of the ants (*Brady et al., 2006*; *Moreau et al., 2006*) for 54 genera representing 21 subfamilies plus two genera of wasps as outgroups. We also compiled behavioral data on the mode of colony foundation for our exemplar species and tested for correlated evolution between queen thoracic morphology and founding behavior. Our comparisons are drawn from a total of 111 species belonging to 93 genera within Formicidae, representing 20 of the 21 extant subfamilies plus the fossil taxon Sphecomyrminae[†] among the four extinct subfamilies.

## The unique thoracic architecture of worker ants

We assessed the relative sizes and configuration of the dorsal plates that form the thoracic exoskeleton in 265 queens and workers belonging to 11 species in five major ant subfamilies (*Table 1*). For each caste of each species, we measured the length of T1, T2, and T3 of 5–17 individuals from museum collections. Our morphometric analyses showed that in ant queens, both T1 and T3 are reduced relative to T2, which makes up most of the thorax (*Figure 1A*). This conforms to the typical proportions in insects where flight is powered exclusively by large wing muscles inside T2 (*Snodgrass, 1935*; *Dudley, 2002*) (e.g., Diptera, Hymenoptera, and Lepidoptera). In contrast, in ant workers, T1 is greatly enlarged and forms a significant portion of the thorax, while T2 is reduced (illustrative SEM image in *Figure 2*). T3 is absent dorsally in workers of most species but, when T3 is distinguishable, the T3/T2 ratio does not differ between castes. In contrast, the ratio between T1 and T2 clearly discriminates workers and

**Table 1.** Ant species studied for morphometrics and/or internal anatomy

| | | Morphometrics | | Dissections | |
|---|---|---|---|---|---|
| **Subfamily** | **Species** | **q** | **w** | **q** | **w** |
| Amblyoponinae | *Amblyopone australis* | 6 | 8 | 2 | 2 |
| Dolichoderinae | *Tapinoma simrothi* | – | – | 6 | 10 |
| Ectatomminae | *Ectatomma ruidum* | – | – | 3 | 5 |
| Formicidae | *Lasius niger* | 15 | 15 | 2 | 4 |
| | *Oecophylla smaragdina* | – | – | 2 | 5 |
| | *Polyrhachis laboriosa* | 13 | 5 | 3 | 8 |
| Myrmeciinae | *Myrmecia simillima* | – | – | 2 | 4 |
| | *Nothomyrmecia macrops* | – | – | 1 | 4 |
| Myrmicinae | *Carebara vidua* | 5 | 3 | 1 | 1 |
| | *Cataulacus wasmanni* | 15 | 15 | 3 | 3 |
| | *Leptothorax pergandei* | 13 | 15 | 1 | 3 |
| | *Messor barbarus* | – | – | 3 | 8 |
| | *Monomorium pharaonis* | 15 | 15 | 2 | 4 |
| | *Monomorium subopacum* | – | – | 2 | 3 |
| | *Pogonomyrmex barbatus* | 15 | 17 | 4 | 5 |
| Ponerinae | *Brachyponera lutea* | 15 | 15 | 3 | 5 |
| | *Harpegnathos saltator* | – | – | 2 | 4 |
| | *Neoponera apicalis* | 7 | 12 | 4 | 10 |
| Pseudomyrmecinae | *Tetraponera aethiops* | 11 | 15 | 4 | 6 |

q = number of queens examined; w = number of workers examined. Generic placement of *Brachyponera lutea* and *Neoponera apicalis* reflects the new reclassification of species within the former paraphyletic genus *Pachycondyla* (Schmidt CA, Shattuck SO, The higher classification of the ant subfamily Ponerinae [Hymenoptera: Formicidae], with a review of ponerine ecology and behavior. Under review).

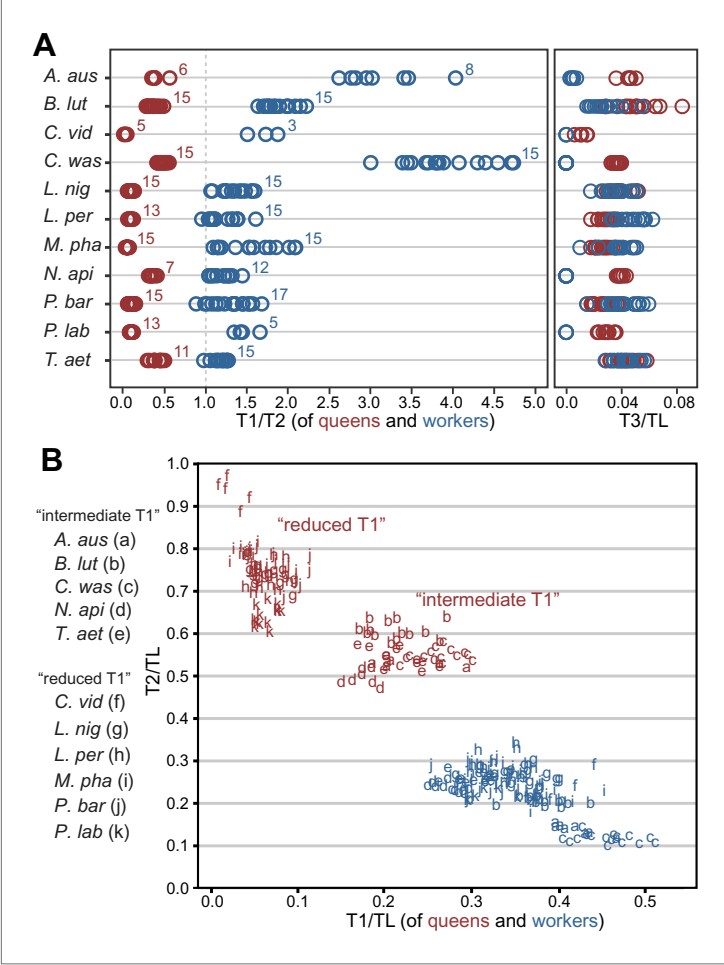

**Figure 1**. Variation in length of first (T1) and second (T2) thoracic segments in ants shows characteristic differences depending on caste and species. (**A**) Relative lengths of T1 and T2 (left) show clear differences between queens and workers for 11 ant species. T3 (right) constitutes a small portion of the total length of the thorax in both queens and workers and, when present (when T3/TL > 0.0), is indistinguishable between castes. Numbers correspond to sample sizes and are equal for both panels (**Table 1**). (**B**) Gradient of investment in neck strength vs flight/storage musculature sorts individuals into three categories. Queens fall into two discrete categories based on the relative lengths of T1 and T2. While the use of T1/T2 in (**A**) emphasizes the distinction between workers and queens and within species variation, T1/TL and T2/TL in (**B**) enables the distinction between queen types across species with large differences in body size. Measurements and ratios are available in the Dryad data repository under DOI doi: 10.5061/dryad.d62p2/1 (**Keller et al., 2014**). Species codes: *A. aus = Amblyopone australis*; *B. lut = Brachyponera lutea*; *C. vid = Carebara vidua*; *C. was = Cataulacus wasmanni*; *L. nig = Lasius niger*; *L. per = Leptothorax pergandei*; *M. pha = Monomorium pharaonis*; *N. api = Neoponera apicalis*; *P. bar = Pogonomyrmex barbatus*; *P. lab = Polyrhachis laboriosa*; *T. aet = Tetraponera aethiops*.

The following figure supplements are available for figure 1:

**Figure supplement 1**. Measurements used in this study.

**Figure supplement 2**. Differences in length proportion of thoracic segments among castes in nine representative species from different subfamilies.

queens. Rather than just showing an overall reduction in T2, consistent with their lack of wings, worker ants have a T1/T2 ratio reversed in relation to queens (**Figure 1A**; SEM images in **Figure 2**). The difference between castes in this ratio depends on the species (Linear model: interaction *Species* x *Caste*, df = 10, F= 68.3, p<0.00001) but it is always greater in workers than in queens (Linear Model: holding the factor *Species* constant, factor *Caste*, df = 1, F= 8975.3, p<0.00001). Visual inspection of

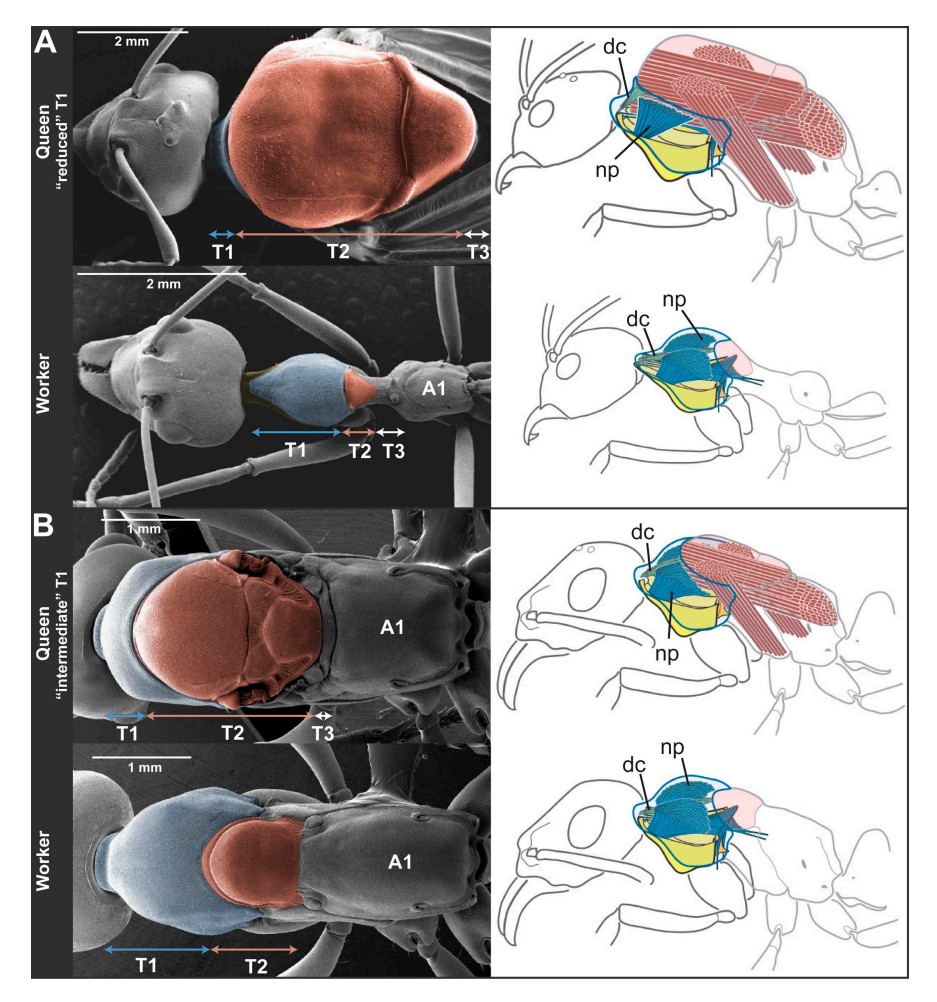

**Figure 2**. Skeletomuscular specialization of queens and workers in ants. The dorsal plate of T1 is always enlarged in workers relative to queens (left column; multiple individuals from 52 genera examined, *Table 2*). Queens can either (**A**) have a reduced T1 and huge T2-associated wing muscles (represented here by *Oecophylla smaragdina*), or (**B**) show a slightly enlarged T1 and associated neck muscles (represented here by *Neoponera apicalis*). T1, T2, and T3, first, second and third thoracic segments respectively; A1, first abdominal segment. Workers of *N. apicalis* lack a discernible T3. Internally (right column), the wing muscles in queens (red) fill most of the thoracic cavity, while the T1 muscles (blue) are narrow and close to the thoracic wall. In all workers examined (see *Table 1* for list of species and sample sizes), the T1 notopleural muscles (np, dark blue) that support the anteroventral plates (yellow) fill the anterior portion of the cavity. The dorsal cervical muscles (dc, light blue; see also *Figure 2—figure supplement 1B*) that in winged queens originate at the anterior phragma and pull the head up at contraction, show a shifted position in workers. In the absence of phragma, these muscles originate at the dorsal boundary between T1 and T2. Rather than being short and thin, they form long and thick bundles that stretch the entire length of the enlarged T1 cavity to their place of insertion on the back of the head (*Figure 2—figure supplement 2*). *Figure 1—supplement 1* has photos of many more species of 'reduced T1' and 'intermediate T1' species for comparison of external thoracic morphology.

The following figure supplements are available for figure 2:

**Figure supplement 1**. Thoracic musculature in queen and worker ants.

**Figure supplement 2**. Internal anatomical adaptations in ant workers for powerful head movement.

an extended sample of species (*Table 2*) from 21 of the 25 ant subfamilies (including the extinct taxon Sphecomyrminae[†]) confirmed the universality of these relative length differences. Castes of all species with specimens available show a differential investment in the growth of T1 and T2. T2 was larger than T1 in queens of all 52 species examined, and T1 was larger than T2 in workers of all 111 species examined (*Table 2*; examples in *Figure 1—figure supplement 2*).

To infer the functional significance of the caste-specific external thoracic configurations, we performed a comparative analysis of the internal skeletomuscular system in queens and workers. We dissected 144 individuals from 19 species belonging to eight subfamilies (*Table 1*) and analyzed both muscle (extent of attachment) and skeletal elements. Our dissections showed that the length of the thoracic segments in dorsal view is a reflection of the volume of the muscles associated with each segment. In the same way that the large T2 of queens is indicative of the presence of large wing muscles, the large T1 in workers reflects the enlargement of muscles in this segment (*Figure 2*, *Figure 2—figure supplement 1*, *Figure 2—figure supplement 2*). Studies in other insects established that homologous T1-associated skeletomuscular elements are involved in the head-thorax articulation or neck (*Snodgrass, 1935, 1956*; *Hartenstein, 2006*). In queens of all 19 ant species dissected (*Table 1*), the neck-associated muscles were short and thin, traversing the narrow space of T1 between the head and the anterior phragma (cuticular invagination) of T2 where the wing muscles attach (*Figure 2*, *Figure 2—figure supplement 1A*). This configuration of neck elements is similar to that of female honey bees irrespective of caste (*Snodgrass, 1956*), and *Drosophila* fruit flies (*Hartenstein, 2006*; *McQuilton et al., 2012*). In contrast, in ant workers, the expansion of T1 and the lack of both anterior phragma and wing muscles result in a larger anterior cavity that contains neck muscles and skeletal pieces in a unique configuration.

The most striking muscular difference between ant castes concerns one of the notopleural pairs of muscles that originate dorsally on T1 (np in *Figure 2*, right column; *Figure 2—figure supplement 1C–H*). The main function of homologous muscles in honey bees is to carry the plates that support the head and serve to move it sideways or rotate it (*Snodgrass, 1956*). In ant queens, where most of the thoracic cavity is filled by the wing muscles (as is the case in all castes of honey bees), these muscles are narrow and close to the thoracic wall. Our dissections revealed that the equivalent muscles in ant workers are hypertrophied, and fill the wider T1 cavity completely. Ant workers also show important differences in internal skeleton associated with T1 (*Figure 2—figure supplement 2B*). This skeletomuscular configuration highlights the increased strength of the workers' neck that powers head movements.

## Two types of queen thoracic architecture

Even though queens invest mostly in the thoracic segment used for flight (T2), our morphometric data showed that queens of different species fall into two discrete categories based on the relative investment into T1. When plotting the normalized length of T1 vs T2 for 130 queens measured (*Table 1*), we can discriminate two clusters of species (*Figure 1B*; where workers of all species form a third cluster). For five of the 11 species in *Table 1*, queens have a reduced T1, almost not visible in dorsal view (*Figure 2A*). The other six species form a category with queens having an intermediate T1, corresponding to enlarged T1 muscles (*Figure 2B*).

To investigate the evolution of queen thoracic configurations across the ant phylogeny, we focused on a total of 54 ant species (those in *Table 2* for which queens were available for measurements) representing 21 subfamilies, as well as two species of wasps from different families as outgroups (*Table 3*). Queens were scored as belonging to the categories 'reduced T1' (22 species, all with T1/T2 < 0.14) or 'intermediate T1' (32 species, all with T1/T2 > 0.28), as seen in *Figure 1B*. This information was combined with a well-established ant phylogeny (*Brady et al., 2006*; *Moreau et al., 2006*) and we used parsimony and maximum likelihood (ML) methods to reconstruct ancestral character states (*Figure 3*). Our analysis showed that an 'intermediate T1' in queens arose in the common ancestor to all ants (ML proportional likelihood = 0.800), and that there were multiple transitions to the 'reduced T1' (*Figure 3*). This reduction seems to have evolved convergently in at least four major ant lineages. Transitions back to an 'intermediate T1' are rare and more recent events, being restricted to the genera *Polyergus* within subfamily Formicinae, and *Cataulacus* and *Metapone* within subfamily Myrmicinae. In contrast, the universal occurrence of a hypertrophied T1 in workers (including the primitive fossil *Sphecomyrma*[†]) supports a single origin of this novel thoracic configuration in the common ancestor of all ants (*Figure 3*).

**Table 2.** List of species surveyed for relative length of thoracic segments

| FAMILY/subfamily | species | queen | | worker | |
|---|---|---|---|---|---|
| | | Museum | Voucher code | Museum | Voucher code |
| FORMICIDAE | | | | | |
| Aenictinae | *Aenictus vaucheri/binghami* | MSNG | CASENT0903754 | AMNH | RAK0094 |
| Agroecomyrmecinae | *Tatuidris tatusia* | DADC | CASENT0178881 | BMNH | RAK0001 |
| Amblyoponinae | *Adetomyrma* sp | | | AMNH | RAK0003 |
| Amblyoponinae | *Amblyopone australis* | ANIC | CASENT0172213 | AMNH | RAK0005 |
| Amblyoponinae | *Amblyopone mercovichi* | | | MCZ | RAK0006 |
| Amblyoponinae | *Apomyrma stygia* | MNHN | CASENT0101445 | MCZ | RAK0083 |
| Amblyoponinae | *Concoctio concenta* | | | MCZ | RAK0011 |
| Amblyoponinae | *Myopopone castanea* | ANIC | CASENT0172069 | AMNH | RAK0012 |
| Amblyoponinae | *Mystrium* sp | CASC | CASENT0104559 | CASC | CASENT0076622 |
| Amblyoponinae | *Onychomyrmex doddi* | | | AMNH | RAK0014 |
| Amblyoponinae | *Prionopelta punctulata* | ANIC | CASENT0172312 | AMNH | RAK0016 |
| Amblyoponinae | *Stigmatomma armigera* | | | AMNH | RAK0004 |
| Amblyoponinae | *Stigmatomma pallipes* | ABS | CASENT0103553 | MCZ | RAK0009 |
| Amblyoponinae | *Stigmatomma pluto* | | | MCZ | RAK0010 |
| Amblyoponinae | *Xymmer muticus* | | | MCZ | RAK0007 |
| Aneuretinae | *Aneuretus simoni* | ANIC | CASENT0172259 | MCZ | RAK0074 |
| Cerapachyinae | *Acanthostichus serratulus* | | | AMNH | RAK0095 |
| Cerapachyinae | *Cerapachys nitidulus* | RAKC | RAK127 | AMNH | RAK0096 |
| Cerapachyinae | *Cerapachys doryloides* | | | AMNH | RAK0097 |
| Cerapachyinae | *Cylindromyrmex brevitarsus* | JTLC | CASENT0610653 | AMNH | RAK0098 |
| Cerapachyinae | *Simopone schoutedeni* | | | AMNH | RAK0099 |
| Dolichoderinae | *Dolichoderus bispinosus* | ALWC | CASENT0173835 | ALWC | CASENT0173833 |
| Dolichoderinae | *Iridomyrmex lividus* | ANIC | CASENT0172066 | ANIC | CASENT0172041 |
| Dolichoderinae | *Leptomyrmex pallens* | | | AMNH | RAK0075 |
| Dolichoderinae | *Tapinoma erraticum* | CASC | CASENT0173200 | AMNH | RAK0078 |
| Dolichoderinae | *Technomyrmex albipes* | CASC | CASENT0060419 | AMNH | RAK0079 |
| Dorylinae | *Dorylus conradti/helvolus* | MSNG | CASENT0903712 | AMNH | RAK0100 |
| Ecitoninae | *Cheliomyrmex morosus* | | | AMNH | RAK0101 |
| Ecitoninae | *Eciton hamatum* | JTLC | INBIOCRI001283500 | AMNH | RAK0103 |
| Ecitoninae | *Labidus coecus* | | | AMNH | RAK0102 |
| Ectatomminae | *Ectatomma tuberculatum* | JTLC | JTLC000014186 | AMNH | RAK0017 |
| Ectatomminae | *Gnamptogenys annulata* | | | AMNH | RAK0018 |
| Ectatomminae | *Gnamptogenys striatula* | MIZA | CASENT0178660 | AMNH | RAK0019 |
| Ectatomminae | *Gnamptogenys bufonis* | | | MCZ | RAK0020 |
| Ectatomminae | *Gnamptogenys minuta* | | | MCZ | RAK0021 |
| Ectatomminae | *Rhytidoponera metallica* | ANIC | CASENT0172346 | ANIC | CASENT0172345 |
| Ectatomminae | *Typhlomyrmex pusillus* | MIZA | CASENT0178662 | AMNH | RAK0023 |
| Ectatomminae | *Typhlomyrmex rogenhoferi* | | | AMNH | RAK0024 |
| Formicinae | *Formica* sp. (*fusca* group) | CASC | CASENT0173171 | AMNH | RAK0080 |
| Formicinae | *Lasius flavus* | CASC | CASENT0173149 | UCDC | CASENT0005406 |
| Formicinae | *Oecophylla smaragdina* | CASC | CASENT0173644 | AMNH | RAK0082 |

*Table 2. Continued on next page*

Keller *et al.* eLife 2014;3:e01539. DOI: 10.7554/eLife.01539

*Table 2. Continued*

| FAMILY/subfamily | species | queen | | worker | |
|---|---|---|---|---|---|
| | | Museum | Voucher code | Museum | Voucher code |
| Formicinae | *Polyergus* sp | RAKC | RAK0129 | RAKC | RAK0130 |
| Formicinae | *Polyrhachis revoili* | CASC | CASENT0403971 | CASC | CASENT0227558 |
| Heteroponerinae | *Acanthoponera minor* | | | AMNH | RAK0025 |
| Heteroponerinae | *Heteroponera brouni* | MCZ | RAK0128 | AMNH | RAK0026 |
| Heteroponerinae | *Heteroponera relicta* | | | AMNH | RAK0027 |
| Leptanillinae | *Leptanilla swani* | AMNH | RAK129 | AMNH | RAK0084 |
| Leptanilloidinae | *Leptanilloides erinys/ biconstricta* | UCDC | CASENT0234616 | AMNH | RAK0104 |
| Martialinae | *Martialis heureka* | | | MZSP | CASENT0106181 |
| Myrmeciinae | *Myrmecia gulosa* | CASC | CASENT0103309 | CASC | CASENT0103310 |
| Myrmeciinae | *Nothomyrmecia macrops* | | | AMNH | RAK0086 |
| Myrmicinae | *Aphaenogaster fulva* | CASC | CASENT0104857 | CASC | CASENT0103585 |
| Myrmicinae | *Carebara vidua* | CASC | CASENT0260121 | CASC | CASENT0010803 |
| Myrmicinae | *Cataulacus wasmanni* | CASC | CASENT0498338 | CASC | CASENT0498558 |
| Myrmicinae | *Leptothorax pergandei* | MCZ | RAK0125 | MCZ | RAK0126 |
| Myrmicinae | *Manica rubida* | | | AMNH | RAK0090 |
| Myrmicinae | *Messor barbarus* | RAKC | RAK0123 | RAKC | RAK0124 |
| Myrmicinae | *Metapone madagascarica* | CASC | CASENT0004524 | MCZ | RAK0093 |
| Myrmicinae | *Monomorium pharaonis* | ABS | CASENT0104094 | ABS | CASENT0104095 |
| Myrmicinae | *Myrmica wheeleri* | MCZ | CASENT0102860 | MCZ | CASENT0102862 |
| Myrmicinae | *Pogonomyrmex uruguayensis* | RAJC | CASENT0172689 | RAJC | CASENT0103054 |
| Paraponerinae | *Paraponera clavata* | RAKC | RAK0122 | AMNH | RAK0028 |
| Ponerinae | *Anochetus mayri* | ABS | CASENT0103555 | MCZ | CASENT0003324 |
| Ponerinae | *Asphinctopone silvestrii* | | | MCZ | RAK0031 |
| Ponerinae | *Belonopelta deletrix* | | | MCZ | RAK0032 |
| Ponerinae | *Bothroponera pachyderma* | | | AMNH | RAK0054 |
| Ponerinae | *Brachyponera croceicornis* | | | AMNH | RAK0051 |
| Ponerinae | *Centromyrmex brachycola* | UCDC | CASENT0178343 | AMNH | RAK0033 |
| Ponerinae | *Cryptopone gilva* | CASC | CASENT0006055 | AMNH | RAK0034 |
| Ponerinae | *Diacamma ceylonense* | | | AMNH | RAK0035 |
| Ponerinae | *Dinoponera lucida* | | | AMNH | RAK0036 |
| Ponerinae | *Dolioponera fustigera* | | | MCZ | RAK0037 |
| Ponerinae | *Emeryopone buttelreepeni* | | | MCZ | RAK0038 |
| Ponerinae | *Hagensia marleyi* | | | MCZ | RAK0053 |
| Ponerinae | *Harpegnathos saltator* | | | AMNH | RAK0039 |
| Ponerinae | *Hypoponera* sp1. | | | AMNH | RAK0040 |
| Ponerinae | *Leptogenys (Leptogenys)* sp.1 | | | AMNH | RAK0041 |
| Ponerinae | *Leptogenys (Lobopelta)* sp.2 | | | AMNH | RAK0042 |
| Ponerinae | *Leptogenys podenzanai* | | | MCZ | RAK0043 |
| Ponerinae | *Loboponera obeliscata* | | | AMNH | RAK0044 |
| Ponerinae | *Loboponera vigilans* | | | AMNH | RAK0045 |
| Ponerinae | *Myopias chapmani* | ANIC | CASENT0172094 | ANIC | CASENT0172093 |

*Table 2. Continued on next page*

*Table 2. Continued*

| FAMILY/subfamily | species | queen | | worker | |
|---|---|---|---|---|---|
| | | Museum | Voucher code | Museum | Voucher code |
| Ponerinae | *Neoponera apicalis* | ALWC | CASENT0103060 | AMNH | RAK0048 |
| Ponerinae | *Neoponera villosa* | | | AMNH | RAK0058 |
| Ponerinae | *Odontomachus bauri* | CASC | CASENT0172630 | AMNH | RAK0030 |
| Ponerinae | *Odontoponera transversa* | BMNH | CASENT0900664 | AMNH | RAK0047 |
| Ponerinae | *Ophthalmopone berthoudi* | | | MCZ | RAK0049 |
| Ponerinae | *Pachycondyla crassinoda* | | | AMNH | RAK0050 |
| Ponerinae | *Cryptopone guianensis* | | | MCZ | RAK0052 |
| Ponerinae | *Pseudoneoponera porcata* | | | AMNH | RAK0055 |
| Ponerinae | *Pseudoponera stigma* | | | AMNH | RAK0056 |
| Ponerinae | *Paltothyreus tarsatus* | | | AMNH | RAK0057 |
| Ponerinae | *Phrynoponera gabonensis* | | | AMNH | RAK0059 |
| Ponerinae | *Platythyrea punctata* | ABS | CASENT0104429 | AMNH | RAK0060 |
| Ponerinae | *Platythyrea turneri* | | | MCZ | RAK0061 |
| Ponerinae | *Plectroctena strigosa* | | | AMNH | RAK0062 |
| Ponerinae | *Ponera alpha* | | | MCZ | RAK0063 |
| Ponerinae | *Ponera pennsylvanica* | CASC | CASENT0006086 | AMNH | RAK0064 |
| Ponerinae | *Psalidomyrmex procerus* | | | AMNH | RAK0065 |
| Ponerinae | *Simopelta oculata* | | | MCZ | RAK0066 |
| Ponerinae | *Streblognathus peetersi* | | | AMNH | RAK0067 |
| Ponerinae | *Thaumatomyrmex atrox* | | | AMNH | RAK0068 |
| Proceratiinae | *Discothyrea oculata* | | | AMNH | RAK0069 |
| Proceratiinae | *Discothyrea testacea* | ABS | CASENT0103848 | AMNH | RAK0070 |
| Proceratiinae | *Proceratium croceum* | ABS | CASENT0104440 | AMNH | RAK0071 |
| Proceratiinae | *Proceratium pergandei* | | | AMNH | RAK0072 |
| Proceratiinae | *Probolomyrmex guineensis* | | | AMNH | RAK0073 |
| Pseudomyrmecinae | *Pseudomyrmex gracilis* | ABS | CASENT0103779 | AMNH | RAK0087 |
| Pseudomyrmecinae | *Tetraponera aethiops* | | | AMNH | RAK0088 |
| Pseudomyrmecinae | *Tetraponera attenuata* | CASC | CASENT0217587 | AMNH | RAK0089 |
| Sphecomyrminae[†] | *Sphecomyrma freyi*[†] | | | AMNH | AMNH NJ-943 |
| SCOLIIDAE | *Scolia nobilitata* | AMNH | RAK0121 | | |
| VESPIDAE | *Metapolybia cingulata* | AMNH | RAK0120 | | |

Information on museum holdings and voucher codes for queens and workers. ABS, Archbold Biological Station; ALWC, Alexander Wild Collection; AMNH, American Museum of Natural History; ANIC, Australian National Insect Collection; BMNH, British Museum of Natural History; CASC, California Academy of Science; DADC, David A. Donoso Collection; JTLC, Jack Longino Collection; MCZ, Museum of Comparative Zoology (Harvard); MIZA, Museo del Instituto de. Zoología Agrícola (Venezuela); MNHN, Muséum national d'Histoire naturelle; MSNG, Natural History Museum, Genoa; MZSP, Museu de Zoologia Universidade de São Paulo; RAJC, Robert Johnson Collection; RAKC, Roberto Keller Collection; UCDC; University of California Davis.
† denotes extinct taxa.

## Queen morphology reflects colony foundation strategy

Out of the two morphological categories of queens we identified, one is closer to workers in size of T1 vs T2 (*Figure 1B*). Similarly for behavior, it is known that queens in some species go through a worker-like phase after they mate and shed their wings. In several lineages, lone founding queens regularly forage outside the nest (they are 'non-claustral'), and can hunt and carry large prey to feed the first

**Table 3.** Queen thoracic morphology and type of colony foundation across ants

| Subfamily | Genus | T1/T2 in queens | T1 in queens | Colony founding | References |
|---|---|---|---|---|---|
| Aenictinae | *Aenictus* | 2.742 | intermediate* | fission | (*Gotwald and Cunningham-van Someren, 1976*) |
| Agroeconomyrmecinae | *Tatuidris* | 0.111 | reduced | unknown | |
| Amblyoponinae | *Amblyopone* | 0.382 | intermediate | non-claustral | (*Haskins and Haskins, 1951*) |
| Amblyoponinae | *Apomyrma* | 0.338 | intermediate | unknown | |
| Amblyoponinae | *Myopopone* | 0.453 | intermediate | non-claustral | (*Ito, 2010*) |
| Amblyoponinae | *Mystrium* | 0.454 | intermediate | non-claustral | (*Molet et al., 2009*) |
| Amblyoponinae | *Prionopelta* | 0.514 | intermediate | non-claustral | (*Ito and Billen, 1998*) |
| Aneuretinae | *Aneuretus* | 0.096 | reduced | claustral | (*Wilson et al., 1956*) |
| Cerapachyinae | *Cerapachys* | 0.364 | intermediate | unknown ICF + fission | (*Brown, 1975*) |
| Cerapachyinae | *Cylindromyrmex* | 0.454 | intermediate | non-claustral | (*Delabie and Reis, 2000*) |
| Dolichoderinae | *Dolichoderus* | 0.061 | reduced | unknown | |
| Dolichoderinae | *Iridomyrmex* | 0.071 | reduced | claustral | (*Hölldobler and Carlin, 1985*) |
| Dolichoderinae | *Tapinoma* | 0.111 | reduced | claustral | (*Kannowski, 1959*) |
| Dolichoderinae | *Technomyrmex* | 0.071 | reduced | claustral | (*Yamauchi et al., 1991*) |
| Dorylinae | *Dorylus* | 0.372 | intermediate* | fission | (*Kronauer et al., 2004*) |
| Ecitoninae | *Eciton* | 0.469 | intermediate* | fission | (*Schneirla, 1949*) |
| Ectatomminae | *Ectatomma* | 0.325 | intermediate | non-claustral | (*Dejean and Lachaud, 1992*) |
| Ectatomminae | *Gnamptogenys* | 0.331 | intermediate | non-claustral | (‡) |
| Ectatomminae | *Rhytidoponera* | 0.363 | intermediate | non-claustral | (*Ward, 1981*) |
| Ectatomminae | *Typhlomyrmex* | 0.504 | intermediate | unknown | |
| Formicinae | *Formica* | 0.076 | reduced | claustral | (*Stille, 1996*) |
| Formicinae | *Lasius* | 0.053 | reduced | claustral | (*Stille, 1996*) |
| Formicinae | *Oecophylla* | 0.066 | reduced | claustral | (*Hölldobler and Wilson, 1978*) |
| Formicinae | *Polyergus* | 0.323 | intermediate | non-claustral† | (*Mori et al., 1995*) |
| Formicinae | *Polyrhachis* | 0.072 | reduced | claustral and non-claustral | (*Lenoir and Dejean, 1994*) |
| Heteroponerinae | *Heteroponera* | 0.485 | intermediate | non-claustral | (§) |
| Leptanillinae | *Leptanilla* | 2.685 | intermediate* | fission | (*Masuko, 1990*) |
| Leptanilloidinae | *Leptanilloides* | 3.021 | intermediate* | fission | (*Donoso et al., 2006*) |
| Martialinae | *Martialis* | n/a | unknown | unknown | |
| Myrmeciinae | *Myrmecia* | 0.485 | intermediate | non-claustral | (*Haskins and Haskins, 1950*) |
| Myrmicinae | *Aphaenogaster* | 0.117 | reduced | claustral | (*Lubertazzi, 2012*) |

*Table 3. Continued on next page*

*Table 3. Continued*

| Subfamily | Genus | T1/T2 in queens | T1 in queens | Colony founding | References |
|---|---|---|---|---|---|
| Myrmicinae | *Carebara* | 0.072 | reduced | claustral | (*Robertson and Villet, 1989*) |
| Myrmicinae | *Cataulacus* | 0.494 | intermediate | unknown | |
| Myrmicinae | *Leptothorax* | 0.090 | reduced | claustral | (*Keller and Passera, 1989*) |
| Myrmicinae | *Messor* | 0.110 | reduced | claustral and non-claustral | (*Brown, 1999*) |
| Myrmicinae | *Metapone* | 0.428 | intermediate | unknown | |
| Myrmicinae | *Monomorium* | 0.132 | reduced | claustral | (*Bolton, 1986*) |
| Myrmicinae | *Myrmica* | 0.071 | reduced | claustral and non-claustral | (*Brown and Bonhoeffer, 2003*) |
| Myrmicinae | *Pogonomyrmex* | 0.097 | reduced | claustral and non-claustral | (*Johnson, 2002*) |
| Paraponerinae | *Paraponera* | 0.086 | reduced | non-claustral | (#) |
| Ponerinae | *Anochetus* | 0.367 | intermediate | non-claustral | (*Brown, 1978*) |
| Ponerinae | *Centromyrmex* | 0.493 | intermediate | non-claustral | (*Dejean and Fénéron, 1996*) |
| Ponerinae | *Cryptopone* | 0.533 | intermediate | non-claustral | (*Peeters, 1997*) |
| Ponerinae | *Ponera* | 0.356 | intermediate | non-claustral | (*Kannowski, 1959*) |
| Ponerinae | *Myopias* | 0.282 | intermediate | non-claustral | (*Peeters, 1997*) |
| Ponerinae | *Odontomachus* | 0.411 | intermediate | non-claustral | (*Brown, 1976*) |
| Ponerinae | *Odontoponera* | 0.524 | intermediate | non-claustral | (*Peeters, 1997*) |
| Ponerinae | *Pachycondyla* | 0.385 | intermediate | non-claustral | (*Peeters, 1997*) |
| Ponerinae | *Platythyrea* | 0.417 | intermediate | non-claustral | (*Peeters, 1997*) |
| Proceratiinae | *Discothyrea* | 0.093 | reduced | non-claustral and claustral | (*Dejean and Dejean, 1998*) |
| Proceratiinae | *Proceratium* | 0.095 | reduced | non-claustral | (¶) |
| Pseudomyrmecinae | *Pseudomyrmex* | 0.479 | intermediate | non-claustral | (**) |
| Pseudomyrmecinae | *Tetraponera* | 0.558 | intermediate | non-claustral | (**) |
| Sphecomyrminae[†] | *Sphecomyrma*[†] | n/a | unknown | unknown | |
| OUTGROUPS | | | | | |
| Scoliinae | *Scolia* | 0.087 | reduced | non-social | (††) |
| Polistinae | *Metapolybia* | 0.074 | reduced | fission | (††) |

Queen thoracic morphology and type of colony foundation across ants. The wasp taxa *Scolia* and *Metapolybia* are included as outgroups.

*species with wingless queens. [†] denotes extinct taxa.

†*Polyergus* is an obligatory social parasite of *Formica* spp.

‡John Lattke, personal communication.

§Rodrigo Feitosa, personal communication.

#Haskins CP, Enzmann EV (1937) Studies of certain sociological and physiological features in the Formicidae. Ann NY Acad Scien 37:97-162; Michael Breed, personal communication.

¶Fuminori Ito, personal communication; Keiichi Masuko, personal communication.

**Philip Ward, personal communication.

††James M Carpenter, personal communication.

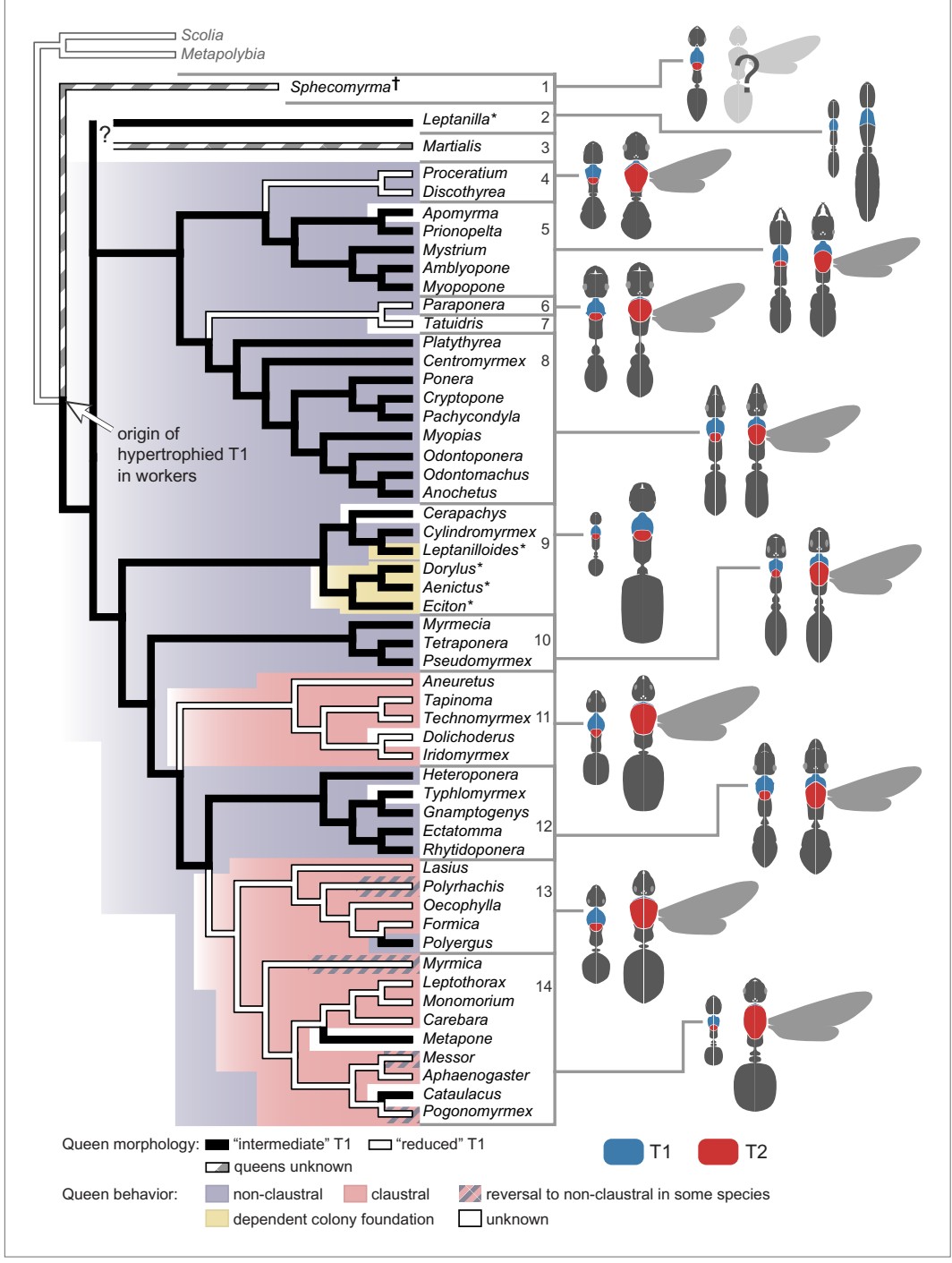

**Figure 3**. Phylogenetic reconstruction reveals a single origin of a hypertrophied T1 in workers and multiple independent origins of 'reduced' T1 in queens. The latter is associated with modifications in modes of colony foundation. Tree branches and tree background are colored for queen morphology and founding behavior respectively, according to the parsimony ancestral reconstruction. Typical queen–worker dimorphism shown to the right to illustrate ratio T1/T2 (not to scale). Species with wingless queens are marked with an asterisk. Phylogeny was pruned from ***Moreau et al. (2006)***. Placement of *Sphecomyrma*† and *Martialis* after ***Grimaldi et al. (1997)*** and ***Rabeling et al. (2008)***, respectively. *Metapolybia* and *Scolia* wasps are included as outgroups. Data on the species are analyzed, and their morphology and type of colony founding behavior are summarized in ***Table 3***. Numbers correspond to major taxonomic groups within Formicidae after ***Ward (2007)***: 1, Sphecomyrminae†; 2, Leptanillinae; 3, Martialinae; 4, Proceratiinae; 5, Amblyoponinae; 6, Paraponerinae; 7, Agroecomyrmecinae; 8, Ponerinae; 9, dorylomorphs; 10, myrmeciomorphs; 11, dolichoderomorphs; 12, ectaheteromorphs; 13, Formicinae; 14, Myrmicinae.

generation of worker larvae (*Haskins, 1970*; *Peeters, 1997*). During several weeks, these non-claustral queens behave much like workers. This contrasts with the vast majority of ants, where founding queens are confined to the nest (they are 'claustral') and, instead of foraging, use their metabolic reserves to feed their first brood (*Hölldobler and Wilson, 1990*; *Wheeler and Buck, 1996*).

To test the hypothesis that the morphological classes associate with the behavioral classes, we compiled data on mode of colony foundation for the 54 ant species in our tree (*Figure 3*; *Table 3*). We could find information for a total of 45 species: 25 non-claustral, 15 claustral, and 4 with dependent colony foundation (i.e. colony fission, when queens are never alone). Unfortunately, there are no data for some of the putative early lineages (the fossil *Sphecomyrma* [*Grimaldi et al., 1997*], and the two rare subfamilies Leptanillinae and Martialinae [*Borowiec et al., 2011*]). Using parsimony and ML methods, we established that non-claustral behavior is the most likely ancestral condition (*Figure 3*; ML proportional likelihood = 0.919). Claustral colony foundation has evolved at least twice independently, with reversals to non-claustral foundation occurring sporadically within some genera. Our reconstruction supports colony fission as a secondary shift among ants (*Cronin et al., 2013*).

Next, we performed a Concentrated Changes test (*Maddison, 1990*) to investigate the phylogenetic correlation between queen thoracic morphology and founding behavior. We found strong support for correlated evolution (p=0.027, calculated by simulation of 100,000 actual changes with two gains and four loses): all queens with an 'intermediate T1' are non-claustral founders, whereas two of four independent origins of queens with a 'reduced T1' coincide with shifts to claustral foundation (*Figure 3*; clades 11 and 13+14). A reversal in morphology to 'intermediate T1' corresponds to a modified claustral behavior in *Polyergus* (clade 13) which parasitizes colonies of *Formica*, hence *Polyergus* queens need to fight to invade the host colonies (*Trager, 2013*). However, sporadic reversals to non-claustral founding have been reported for a few species (*Lenoir and Dejean, 1994*; *Brown, 1999*; *Johnson, 2002*; *Brown and Bonhoeffer, 2003*) that according to our morphological survey are not accompanied by reversals in queen morphology (*Figure 3*). Modeling suggests that such facultative reversals to non-claustral behavior are likely to occur in cases of increased resource availability (*Johnson, 2002*; *Brown and Bonhoeffer, 2003*). We did not observe changes in the T1/T2 ratio in lineages that secondarily evolved colony fission, even though this mode of colony foundation is known to co-occur with wing-loss in queens (*Cronin et al., 2013*). This suggests that, despite being wingless, in the absence of the selective pressures related to worker-like foraging (as in non-claustral queens) or of the need for storing metabolic reserves as flight muscles (as in claustral queens), queens in those lineages maintain the ancestral T1/T2 ratio (e.g. dorylomorph clade in *Figure 3*).

## Discussion

The ecological dominance of ants in terrestrial ecosystems is unparalleled in the animal kingdom (*Wilson, 1971*; *Folgarait, 1998*). Because no other group of social insects reaches equivalent levels of adaptive radiation and species-richness (*Hölldobler and Wilson, 1990*), it seems that factors in addition to social behavior and division of labor promoted ant diversification. The evolutionary success of ants is indisputably associated with a strong divergence between queens and workers. A caste of flightless workers specialized in non-reproductive activities is unique among social Hymenoptera. However, rather than being just simplified, wingless versions of the queen, the thorax of ant workers has its own specialization. Relative to the thoracic morphology of queens, which is typical of species of flying insects, worker ants have an unusually large T1 and T1-associated muscles, which provide superior strength and mobility to the neck controlling head movements.

Control of the head is of great importance for ant workers, which in some species singly hunt and carry prey up to 30-90 times their weight (*Dejean, 2011*; *Dejean et al., 2012*). Among insects, ant carrying behavior is unique in that workers lift their load off the ground. Many other insects can move relatively large objects, but by dragging (e.g., spider wasps) or rolling them (e.g., dung beetles) on the ground, or holding them while flying (e.g., robber flies). Biomechanical studies on grass-cutting ants have shown that workers perform controlled head movements at the neck articulation when transporting large objects (*Moll et al., 2010*). Precise head movements are essential to reduce displacement of the center of mass, and retain stability while carrying objects many times the workers' weight and length. Our finding that worker ants differ from queens and other flying insects in the configuration and size of the T1-associated muscles suggests that ants can achieve this biomechanical feat by virtue of their specialized neck musculature. This represents a striking structural innovation, differentiating ant workers from the typical flying insects, which had not been recognized until now. Their distinctive internal

skeletomuscular modifications presumably enhance their behavior as flightless foragers and heavy-load transporters. We propose that the modified T1 was an innovation that helped ants to use their heads and mandibles in novel ways, and hence exploit a broader spectrum of trophic resources. Compared to social bees and wasps (*Hölldobler and Wilson, 1990*), where worker morphology is constrained by the requirements of a winged thorax, mandibular morphology and function have specialized enormously across ant lineages (*Paul and Gronenberg, 1999*), in parallel with their much greater diversification of foraging habits.

Our analysis also showed that queens fall in two distinct anatomical types that evolved in association with the two strategies of independent colony founding. Foraging activity during independent foundation is high in non-claustral species vs absent in claustral species. Non-claustral queens have a T1 that is closer in size to that of workers, while claustral queens, which do not go through a worker-like phase, have a much more reduced T1. Unfortunately, biomechanical data of neck strength in queens are difficult to obtain because they are evasive and, especially in claustral species, cannot be induced to carry objects. Claustral queens have an enlarged T2 relative to non-claustral queens, reflecting the existence of massive wing muscles (*Figure 2A*). A correlation between increased wing muscle mass and claustral behavior has been suggested before: these larger muscles do not function to enhance flight, rather they are a solution for storing amino acids that are essential for feeding the first generation of workers without outside foraging (*Jones et al., 1978*; *Peeters, 2012*) . We specu-late that, during the acquisition of claustral behavior, the decrease in foraging activity lessened the constraint on the size of T1, thus allowing T2 to expand and accommodate larger wing muscles as metabolic reserves. Differences in the nesting habits of queens, such as excavating a nest vs nesting in pre-existing cavities, might also impose variable muscle requirements. However, this type of behavioral differences occurs across species in a scattered pattern that does not match the anatomical categories we revealed. There are examples of nest excavating by queens with 'intermediate' (e.g., *Amblyopone*) and 'reduced' T1 (e.g., *Pogonomyrmex*), and of nesting in pre-existing cavities by 'intermediate' (e.g., *Tetraponera*) as well as 'reduced' T1 (e.g., *Leptothorax*) species.

While data on queen morphology is readily accessible from museum collections for many species, knowledge about their founding behavior remains sparse. There is no published information in many important genera, possibly because this requires field observations of behavior at an appropriate time of the year. Our findings provide a means of predicting colony foundation strategy from the morphology of the queen thorax, and thus guide field research on particular species of interest. For example, within the subfamily Myrmicinae (clade 14 in *Figure 3*), the genera *Cataulacus* and *Metapone* show independent reversals to an 'intermediate T1' in queens, suggesting that colony foundation is not claustral as in closely related genera. Importantly, the phylogenetic component of our correlation provides a powerful tool to infer the ecology of extinct clades for which behavioral observations are impossible. For example, we lack data on queens of two early lineages, the extinct *Sphecomyrma*[†] and the enigmatic *Martialis*, but based on our reconstructions we can predict that they will have an 'intermediate T1' and behave non-claustrally.

Our finding that the ratio of the lengths of T1 and T2 is inverted between queens and workers suggests that a morphological trade-off was at play in determining the relative size of these two segments. It is likely that T1 can become hypertrophied only at the expense of a reduced, non-functional T2. Indeed, our anatomical analysis showed that some of the internal modifications of T1 in workers are only possible in conjunction with a complete absence of wing muscles. Conversely, only queens with a highly reduced T1 have an expanded T2 that constitutes most of the thoracic dorsum (queens with intermediate T1 are also intermediate for T2, see *Figure 1B*). This morphological trade-off between adjacent body segments can occur due to competition for metabolic resources during pre-adult development (*Nijhout and Emlen, 1998*). It is possible that the functional cost of enlarging T2 (reserves for colony founding) at the expense of T1 (reduced neck strength and work performance), occurred when founding behavior gradually shifted to claustral, with a decreased need to forage outside the nest.

## Materials and methods

### Phylogenetic sampling

We compared the thorax of queens and workers across multiple species representing all major ant lineages. First, we measured the length of the thoracic segments and entire thorax in a sample of

individuals belonging to 11 different species from five ant subfamilies (*Table 1*). Second, we used a database of scanning electronic micrographs (*Keller, 2011*) and an online database of light microscopy images (http://www.antweb.org) to further assess the extent of the taxonomic distribution of the traits of interest. We inspected the external thorax of workers belonging to 110 ant species, and of queens belonging to a subset of 47 species where this caste is known or available (listed in *Table 2*). This taxon sampling represents all 21 extant subfamilies with the exception of the dorylomorph subfamily Aenictogitoninae (for which only males have been formally described), and includes the extinct subfamily Sphecomyrminae†. Lastly, we analyzed internal thoracic anatomy by dissecting multiple individuals from 19 representative species (*Table 1*). Our sample of queens included individuals both before and after the phase of muscle reabsorption, as assessed by the shedding of their wings.

## Morphometrics and anatomical analysis

For the quantitative characterization of the thorax, we took dorsal and lateral photographs of pinned specimens (Museum of Comparative Zoology, Harvard University) with a JVC digital camera mounted on a Leica MZ16 binocular microscope (images are deposited in the Dryad data repository under DOI doi: 10.5061/dryad.d62p2/2). We then measured (ImageJ, http://rsb.info.nih.gov/ij) the dorsal length of the first (T1), second (T2), and third (T3) thoracic segments along the midline, and the total thoracic length (TL) as the diagonal length in profile from the anterior-most point of T1 to the posterior-most point of T3 (*Figure 1—figure supplement 1*; measurements are available in the Dryad data repository under http://dx.doi.org/10.5061/dryad.d62p2/1).

For the analysis of internal anatomy we performed muscle preparations using specimens fixed in either 80% ethanol or 4% formaldehyde, and sectioning their thoraces in sagittal and parasagittal planes or disarticulating the plates of the thorax. Muscle preparations were stained in 0.2% methylene blue (Sigma-Aldrich) to increase contrast against other tissues. We also performed skeletal preparations by disarticulating specimens with overnight digestion of soft tissues in 10% KOH, and kept in 90% ethanol for inspection. When necessary (i.e., lightly pigmented specimens), skeletal preparations were stained in 70% ethanol saturated solution of Chlorazol Black E (Sigma-Aldrich).

## Statistical analysis

All analyses were performed with R (*R Development Core Team, 2008*). Residuals of the models have been checked for normality and equality of variance, and data have been transformed when necessary. To compare the relative investment in the thoracic plates 1 and 2 between castes, we performed a linear model (LM) constructed as $\sqrt{(T1/T2)} \sim species * caste$. T1/T2 is the ratio of the thoracic plate 1 length 'T1' over the length of the thoracic plates 2 'T2'. '*' indicated that the effects were tested for both main factors as well as interaction.

## Phylogenetic mapping and correlation of queen behavior and morphology

We scored the queens of our 56 exemplar species as either 'reduced' or 'intermediate'. We divided the length of T1 by the length of T2, and determined a cut-off index equal to 0.25 based on our previous morphometric analysis. We then assigned 'reduced' to queens falling below the cut-off value and 'intermediate' for queens falling above it. Missing data (i.e., the unknown queens of *Sphecomyrma freyi†*, and *Martialis eureka*) (*Grimaldi et al., 1997*; *Rabeling et al., 2008*) were coded as '?'. For the modes of colony foundation we assigned states for 'non-claustral', 'claustral' and 'fission' based on records from the scientific literature (*Table 3*). Unknown mode of colony foundation was coded as '?'. Data in *Table 3* correspond to a single queen for each of the species in *Figure 1* (except *Brachyponera lutea* because its exact phylogenetic position within the subfamily Ponerinae remains undetermined) and from 44 more species (listed in *Table 2*).

Character evolution was reconstructed under parsimony using WinClada (*Nixon, 2002*) and under maximum likelihood (ML) using Mesquite (*Maddison and Maddison, 2012*), under the Mk1 model, (*Lewis, 2001*). Ambiguous optimizations under parsimony were resolved using DELTRAN. This algorithm gave results closer to the ML analysis than did the ACCTRAN parsimony algorithm. Tree topology with branch lengths was pruned from *Moreau et al. (2006)*. We implemented the concentrated changes test (*Maddison, 1990*), using MacClade (*Maddison and Maddison, 2005*), to test for a correlation between modes of colony foundation and queen morphology. This test calculates the probability that changes in a binary character along the phylogeny are distributed randomly on the branches defined by a second binary character. We therefore transformed our data on behavior and morphology to

binary characters by pruning out the branches with fission and wingless queens, since both traits always co-occur in the phylogeny.

## Acknowledgements

We thank Stefan Cover and Gary Alpert for access to the MCZ collection and imaging respectively (Cambridge, USA), and David Duneau for advice on statistical analysis. The comments made by the editor and three reviewers helped to improve our manuscript.

## Additional information

### Funding

| Funder | Grant reference number | Author |
|---|---|---|
| Portuguese Science and Technology Foundation (FCT) | SFRH/BPD/65529/2009 | Roberto A Keller |
| Portuguese Science and Technology Foundation (FCT) | PTDC/BIA-BEC/100243/2008 | Patrícia Beldade |
| Oeiras Municipality Installation Grant | | Patrícia Beldade |

The funders had no role in study design, data collection and interpretation, or the decision to submit the work for publication.

### Author contributions

RAK, Conception and design, Acquisition of data, Analysis and interpretation of data, Drafting or revising the article; CP, Conception and design, Analysis and interpretation of data, Drafting or revising the article, Contributed unpublished essential data or reagents; PB, Analysis and interpretation of data, Drafting or revising the article

## Additional files

### Major dataset

The following dataset was generated:

| Author(s) | Year | Dataset title | Dataset ID and/or URL | Database, license, and accessibility information |
|---|---|---|---|---|
| Keller RA, Peeters C, Beldade P | 2014 | Data from: Evolution of thorax architecture in ant castes highlights trade-off between flight and ground behaviors | http://dx.doi.org/10.5061/dryad.d62p2 | Data available from the Dryad Digital Repository. |

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
