## [Decision Letter]

Thank you for sending your work entitled “Evolution of thorax architecture in ant castes highlights specializations for behaviors on the ground” for consideration at *eLife*. Your article has been favorably evaluated by a Senior editor and 3 reviewers, one of whom is a member of our Board of Reviewing Editors.

The Reviewing editor and the other reviewers discussed their comments before we reached this decision, and the Reviewing editor has assembled the following comments to help you prepare a revised submission.

The paper presents an extensive comparative study of thorax segments in 21 of the 25 extinct and extant ant subfamilies. It shows that major differences can occur between queens and workers and the authors relate this to the ecology of the animals. Using a phylogenetic framework, the authors show how these characters and character switches have evolved.

The work is well presented and the major conclusions are clear and novel. This is a good example of functional polyphenism, which may become a textbook example of how ecology can drive functional adaptations, even within species.

All three reviewers agree that the data are interesting and should be published. However, there is a substantive concern with respect to data presentation that needs to be addressed in a revised version.

The data are presented in a rather streamlined way, which makes one wonder how much room for alternative interpretations might exist. It will be necessary to present the data on the original measurements, ideally all documented with the corresponding pictures of the samples. Part of this is provided, but it is not clearly described as to what exactly has been measured. It would be useful to include a sketch with hymenopteran anatomy pointing out the special structure of the propodeum, and indicating how the thorax and the segment borders were identified.

There is also not enough quantitative information and documentation on the comparative analysis of the internal muscles. More details will be required, complemented with suitable pictures to give an impression on the data on which the drawing is based.

The interpretation of the elongated pronotum of worker ants mainly in terms of the increased strength of the workers' neck should be revisited. As the elongation appears to be associated at least in some species with a taper towards the head (Figure 2), the modification may not only provide space for more powerful muscles, but also enable a wider movement range for the neck joint which may be beneficial for foraging workers.

---

## [Author Response]

*All three reviewers agree that the data are interesting and should be published. However, there is a substantive concern with respect to data presentation that needs to be addressed in a revised version*.

*The data are presented in a rather streamlined way, which makes one wonder how much room for alternative interpretations might exist. It will be necessary to present the data on the original measurements, ideally all documented with the corresponding pictures of the samples*.

We are submitting a spreadsheet with all the original measurements and all 530 images from where the data was gathered.

*Part of this is provided, but it is not clearly described as to what exactly has been measured. It would be useful to include a sketch with hymenopteran anatomy pointing out the special structure of the propodeum, and indicating how the thorax and the segment borders were identified*.

We added a new figure supplement to Figure 1 (Figure 1—figure supplement 1) with images of a queen and a worker where we detail how the measurements were taken and the position of the propodeum with respect to the thoracic segments. In the legend to that figure supplement, we also explain the rationale behind taking some of the lengths in dorsal view and others in lateral view.

*There is also not enough quantitative information and documentation on the comparative analysis of the internal muscles. More details will be required, complemented with suitable pictures to give an impression on the data on which the drawing is based*.

We added an extra figure supplement to Figure 2 (Figure 2—figure supplement 1) with pictures of the dissected muscles from different angles. This new plate supplements the drawings in Figure 2 by showing images with the differences between queens and workers for the relevant muscles.

*The interpretation of the elongated pronotum of worker ants mainly in terms of the increased strength of the workers' neck should be revisited. As the elongation appears to be associated at least in some species with a taper towards the head (*Figure 2*), the modification may not only provide space for more powerful muscles, but also enable a wider movement range for the neck joint which may be beneficial for foraging workers*.

The reviewers make an interesting point. Indeed, a narrow neck joint should enable a wider range of movement of the head. However, this narrow head-to-thorax articulation is a general feature of ants, rather than a specialization unique to workers, as it might seem from Figure 2. In fact, a tapered neck occurs in ants and in other prognathous insects (i.e., those in which the head is positioned so that mouthparts point forwards rather than downwards).

For the species in Figure 2 (*Oecophylla smaragdina*), the neck of the worker seems indeed much more tapered than the neck of the queen. In this species, the queen thoracic morphology is highly divergent having a much reduced T1 (as is characteristic of claustral queens) and an unusually broad neck on a very bulky thorax (even in relation to other claustral queens, which do not have as bulky a thorax). In Figure 1—figure supplement 2, we have images of queens and workers from several claustral and non-claustral species, where readers can see that more tapered necks are not a specific feature of workers. We now refer to this figure supplement in the legend to Figure 2.